

# Observations of stratospheric streamers and frozen-in anticyclones in aerosol extinction

Christian Löns[1], Ronald Eixmann[2], Christine Pohl[3,*], Alexei Rozanov[3], and Christian von Savigny[1]

[1]Institute of Physics, University of Greifswald, 17489 Greifswald, Germany
[2]Leibniz-Institute of Atmospheric Physics at the University of Rostock, 18225 Kühlungsborn, Germany
[3]Institute of Environmental Physics, University of Bremen, 28359 Bremen, Germany
[*]now at: Department of Physics, Lund University, Lund, 22100, Sweden

**Correspondence:** Christian Löns (christian.loens@uni-greifswald.de)

**Abstract.** When the polar vortex meanders and shifts towards the equator, air masses from the tropics and subtropics can be transported towards the pole in so-called tropical-subtropical streamers. These large-scale structures are areas of low potential vorticity and high pressure, containing dry air with a high ozone mixing ratio. The presence of these streamers can also be seen in changes in stratospheric optical properties. Satellite instruments such as OMPS-LP measuring the limb scattering of

these aerosols are capable of observing an increase in the aerosol extinction coefficient in the mid-stratosphere at the edge of the vortex. The high spatial sampling of the limb instrument ensures that the trajectory of the streamer can be accurately monitored. Following a displacement and deformation of the vortex, aerosol transport to high latitudes occurred in the Northern Hemisphere in spring 2017. The additional stratospheric aerosol mass of around 1,000 t at an altitude of 24–38 km remained at middle and high latitudes for just under a month this year. This aerosol mass increase resulted in an estimated 70 % rise

in the total mass within this altitude range at high latitudes. Frozen-in anticyclones, in which low latitude air is trapped in the circulation at high latitudes after the end of the polar vortex, can also be observed in the aerosol extinction coefficient. The observation of a particularly long-lived anticyclone in 2005, which is visible in the aerosol extinction coefficient, is presented. This is the first study documenting streamer events and frozen-in anticyclones in stratospheric aerosols.

## 1 Introduction

The stratospheric part of the polar vortex (hereinafter referred to as the polar vortex) is a large area of cold rotating air. The westerly winds that shape it extend from just above the tropopause into the mesosphere (Waugh et al., 2017). It forms in the autumn months in the polar regions after the onset of the polar night. Here, the atmosphere is not heated by solar radiation. A strong temperature difference arises between the equator and the pole, which creates a pressure gradient with a region of lower pressure at the pole (Schoeberl and Hartmann, 1991). The air flowing towards the pole is deflected eastwards by the Coriolis

force and a circumpolar vortex is formed (Schoeberl and Newman, 2015). The maximum of these intensified westerly winds is a good approximation of the edge of the vortex (Waugh and Polvani, 2010). Within the vortex, wind speeds are minimal. While the Antarctic vortex flows very evenly around the pole, the Arctic polar vortex is more strongly characterized by waves and disturbances (van Loon et al., 1973; Waugh et al., 2017).





So-called sudden stratospheric warmings (SSWs) are an important and frequently occurring zonal indicator of such distur-
bances. Their observation was first published in Scherhag (1952). During SSWs, planetary waves propagate vertically from the
troposphere and transport westward angular momentum upwards (Baldwin and Dunkerton, 2001). These waves often break in
the surf zone in the mid-latitudes of the stratosphere and can lead to a weakening or collapse of the vortex (Matsuno, 1971;
Baldwin et al., 2021). The cold air near the pole descends and the polar temperature in the lower and mid-stratosphere rises
abruptly due to the warmer air coming from above and the associated adiabatic heating. This results in a change in the usually
westerly wind. There is no unambiguous standard definition for SSWs and definitions often differ in the selection of char-
acteristic criteria, threshold values or are defined for a particular latitude or altitude (Butler et al., 2015; Butler and Gerber,
2018). A frequently used and easily applicable definition was proposed by Charlton and Polvani (2007), which is employed in
this study. According to this definition, major SSWs are defined by the reversal of the mean zonal wind at 60° N and 10 hPa.
A distinction often made is between minor SSWs (weakening of the wind) and major SSWs (reversal of the wind direction)
(Labitzke, 1981; Andrews et al., 1987). Major SSWs can be divided into two types: displacement events in which the polar
vortex is deflected and splitting events in which the vortex splits into two or more vortices (Schoeberl and Hartmann, 1991;
Charlton and Polvani, 2007). As shown by Butler et al. (2015), the statistical frequency of SSWs depends strongly on the
respective definition. According to Charlton and Polvani (2007), in the Northern Hemisphere major mid-winter SSWs occur
about six times per decade, minor SSWs about once per year (Gogoi et al., 2023). A final warming marks the end of the polar
vortex season.

In order to understand the transport processes in the winter stratosphere, Plumb (2002) suggests dividing it into three merid-
ional regions: the tropics, the surf zone and the polar vortex. The subtropics largely serve as a transport barrier to isolate the
tropics from the surf zone (Plumb, 1996), as could be shown by the tape recorder signature in the tropical lower stratosphere
(Mote et al., 1996). As described in McIntyre and Palmer (1984), the surf zone is the region into which air masses can be
transported vertically by breaking planetary waves. Similarly, the edge of the polar vortex serves as a transport barrier between
the surf zone and the centre of the polar vortex. It can be characterized by the maximum gradient of potential vorticity (Nash
et al., 1996) or the maximum of the circumpolar westerly winds (Waugh and Polvani, 2010). It is stronger than the subtropical
barrier (Krüger et al., 2005).

Small-scale mixing processes through transport barriers are referred to as filaments. Filaments are finger-like structures with
a vertical extent of less than 2.5 km (Reid and Vaughan, 1991) and a horizontal extent of less than 1000 km (Reid et al., 1993).
Such structures, which were initially found mainly in ozone profiles and are sometimes also referred to as laminae (Dobson,
1973), were explained by Appenzeller and Holton (1997) by differential advection of air masses of different origins. Filaments
occur particularly frequently near the tropopause of the subtropics and mid-latitudes as well as at the edge of the polar vortex
(Dobson, 1973; Reid and Vaughan, 1991; Appenzeller and Holton, 1997).

Sometimes the term 'stratospheric streamer' is used to describe these vertical transport processes from the stratosphere to
the troposphere (Appenzeller and Davies, 1992; Wernli and Sprenger, 2007). In this paper, the term stratospheric streamer
is employed to denote large-scale tongue-like structures that transport tropical-subtropical or polar air masses towards mid-
latitudes in the mid-stratosphere, as described in Offermann et al. (1999) and Krüger et al. (2005).



An early descriptive study of such transport processes was published by Leovy et al. (1985). Using satellite measurements from the Nimbus 7 Limb Infrared Monitor of the Stratosphere (LIMS), tongues of ozone-rich air were observed after the polar vortex at 10 hPa in the Northern Hemisphere was deflected from the pole in early 1979. These ozone-rich tongues, characterized by their spiral trajectory towards the pole, exhibited low potential vorticity. The occurrence of streamers has been shown to be primarily associated with the disturbance of the polar vortex, leading to the edge of the polar vortex reaching low latitudes and tongues of tropical air being drawn towards mid-latitudes (Offermann et al., 1999). It has been demonstrated that this process can occur in both hemispheres during the winter months, although it is less pronounced in the Southern Hemisphere. This is due to the typically smaller amplitude of planetary waves (Randel et al., 1993).

These streamers have a width ranging from 500 to 2000 km, a length exceeding 10,000 to 20,000 km, and persist for a duration of 1 to 3 weeks (Krüger et al., 2005). Observations and model simulations have identified two primary pathways over the Atlantic and over Asia, respectively (Offermann et al., 1999; Krüger et al., 2005). Such meridional transport has been observed for various trace gases, including nitric acid ($HNO_3$) (Offermann et al., 1999), nitrous oxide ($N_2O$) (Randel et al., 1993; Offermann et al., 1999), ozone ($O_3$) (Leovy et al., 1985; Hocke et al., 2017) and water vapour ($H_2O$) (Randel et al., 1993).

Another phenomenon associated with the polar vortex, whereby air from lower latitudes is compressed and can be held at high latitudes for several months, is a frozen-in anticyclone (FrIAC). The first documentation of this phenomenon was by Manney et al. (2006), who observed the FrIAC using MLS data of $N_2O$, $H_2O$ and ozone following the final warming of the Arctic polar vortex in March 2005. The vertical extent of this FrIAC was found to be approximately 25–45 km, and characteristic tropical signals of high $N_2O$ and low $H_2O$ were still identifiable several months later in August 2005. FrIACs were also described for other years, such as in 2003 (Lahoz et al., 2007), 2007 (Thiéblemont et al., 2011) and 2011 (Allen et al., 2013).

The shape of the vortex also influences the global distribution of stratospheric aerosol. Satellite instruments can be used to retrieve the aerosol extinction coefficient, with the two most common measurement methods being limb scattering and solar occultation.

In context of solar occultation the aerosol extinction coefficient is a measure of the attenuation of solar radiation due to its scattering or absorption by particles, such as stratospheric sulfate particles. Solar occultation, in contrast to limb scattering measurements, allows for the retrieval of the extinction coefficient without the necessity of prior assumptions regarding the composition and size of the particles (e.g. McCormick, 1982). In the context of limb scattering the aerosol extinction is predominantly an amplification of the measured signal due to stronger scattering by aerosols. It has the advantage of considerably higher temporal and spatial coverage (e.g. Loughman et al., 2004; von Savigny et al., 2015) and is therefore well suited for global observations of aerosol extinction.

The extinction coefficient above the Junge layer decreases with altitude. The presence of liquid aerosol particles at an altitude of 30 km to 35 km is constrained by the positive temperature gradient in the stratosphere. As the temperature increases, the saturation vapour pressure of $H_2SO_4$ increases by several orders of magnitude. This process has been shown to impede the development of new $H_2SO_4$ particles that are situated above an altitude of around 36 km. The formation of these particles



has been observed to be exclusive to regions where the $H_2SO_4$ partial pressure exceeds the $H_2SO_4$ saturation vapour pressure
(Hamill et al., 1977). Furthermore, an increase in saturation vapour pressure is accompanied by an increase in the rate of evaporation. An increasing extinction coefficient with altitude can therefore be explained either by dynamic effects or by material coming from above, potentially meteoric or space debris.

A decrease in the extinction coefficient with increasing latitude towards the pole is to be expected. This is due to the fact that sulfate aerosols form in the tropical tropopause region and most active volcanoes are also located in the subduction zones
of the tropics and subtropics. Meridional mixing is limited by the dynamic transport barriers in the subtropics and at the edge of the polar vortex.

The present work is based on data products of the Institute of Environmental Physics of the University of Bremen for aerosol extinction derived from limb scattering for the instruments SCIAMACHY V3.0 (von Savigny et al., 2015; Rieger et al., 2018; Sofieva et al., 2024) and OMPS-LP V2.1 (Rozanov et al., 2024), hereafter OMPS-UBr. Aerosol extinction data from
105 2002 to 2023 is available for these two data products. In order to also estimate the size information of the observed aerosols, measurements and derived particle size distribution (PSD) parameters from the Stratospheric Aerosol and Gas Experiment (SAGE) III instrument on the ISS are used.

The transport of tropical aerosol towards mid-latitudes was observed in satellite measurements following the eruption of Mt. Pinatubo in 1991 (McCormick and Veiga, 1992; Trepte et al., 1993). The relationship between the extinction coefficient
and the polar vortex was demonstrated by Thomason et al. (2007) using the potential vorticity (PV) as a tracer for the vortex. Above an altitude of 20 km, a negative correlation between the extinction coefficient and PV is observed, indicating that as PV increases, the extinction coefficient decreases. Therefore, the aerosol extinction within the vortex is minimal. In this study, in addition to the low extinction coefficient within the polar vortex, an increased extinction coefficient is observed at high and medium latitudes at the edge of the vortex.

The aim of the work is to describe anomalies in aerosol extinction coefficient in the high latitudes that are related to a changing or collapsing polar vortex, in particular the meridional transport of aerosol towards the pole in the Northern Hemisphere. The period of spring 2017 stands out because the transport at the edge of the vortex is clearly visible and tropical air remains at middle to high latitudes for a month.

Section 2 describes the data sets of the SCIAMACHY, OMPS-UBr and SAGE III/ISS satellite measurements and the
120 ECMWF ERA5 reanalysis data. Section 3.1 presents results of the zonal mean aerosol extinction coefficient in the Northern Hemisphere for the period of OMPS measurements at different altitudes and latitude bins. Sections 3.2 and 3.4 describe the development of the polar vortex and the aerosol extinction coefficient in early 2017 and 2005. In Sect. 3.2 the methodology used to estimate the transported sulfate aerosol mass is explained. The interpretation and validity of the results are discussed in Sections 4 and 5.



## 2 Instruments and Methodology

### 2.1 Limb scattering instruments

The study of aerosol extinction associated with the polar vortex requires high spatial and temporal resolution. Therefore, the focus of this work is the investigation of stratospheric aerosol data sets obtained from remote sensing with a high spatial sampling, such as limb scattering instruments or atmospheric lidars. However, spaceborne lidars are not well-suited to the investigation of background conditions in the mid-stratosphere in this study due to relatively low signal-to-noise ratios. Instruments in limb viewing geometry measure the intensity of limb-scattered solar radiation, by repeating line-of-sight measurements at different tangent height altitudes from approximately 0 to 100 km, or simultaneously over all heights. Limb scattering instruments are confronted with the challenge of receiving both reflected and scattered light from the Earth and atmosphere along its line of sight. This complicates the methodology, with each measurement depending not only on the aerosol extinction but also on the conditions of the atmosphere and the aerosol microphysical properties such as aerosol size and composition. In addition, as is usually the case in remote sensing, aerosol extinction becomes increasingly difficult to measure with height due to a decreasing signal-to-noise ratio, and the error in the extinction coefficient increases.

The SCanning Imaging Absorption spectroMeter for Atmospheric ChartographY (SCIAMACHY) instrument was launched on 1 March 2002 on board the European Environmental Satellite (Envisat) into a sun-synchronous orbit at an altitude of approximately 800 km. It was operational from August 2002 to April 2012, during which time it measured, amongst other things, the scattered solar radiation in limb and nadir geometry. A grating spectrometer recorded the radiation in eight wavelength channels in the spectral range 214–2386 nm with a spectral resolution of 0.2–1.5 nm (Bovensmann et al., 1999). The retrieval of aerosol extinction from SCIAMACHY measurements was described in von Savigny et al. (2015) and further developed in Rieger et al. (2018). Level 2 version 3.0 aerosol extinction data from the University of Bremen were used for this work, described in Sofieva et al. (2024).

The Ozone Mapping and Profiler Suite Limb Profiler (OMPS-LP) was first launched on 28 October 2011 on board the Suomi National Polar-orbiting Partnership (S-NPP) satellite as one of three OMPS instruments into a sun-synchronous orbit at an altitude of approximately 800 km. The instrument has been in operation since February 2012 and remains active to the present day. Additionally, OMPS-LP was launched on board the NOAA-21 satellite in December 2022. The OMPS-LP instrument is a prism spectrometer covering a spectral range of 280-1000 nm with an increasingly coarser spectral resolution towards infrared wavelengths (Jaross et al., 2014). OMPS-LP has three slits separated horizontally by 4.25°. The measured tangent points are 250 km apart. The solar radiation scattered by the atmospheric limb is simultaneously recorded from the ground up to an altitude of 80-100 km.

The OMPS-LP data product from the University of Bremen is used for this work. One advantage of this data set is that the aerosol extinction coefficient is usually derived to a greater height than the derived aerosol product of NASA. The retrieval of vertical profiles of aerosol extinction from limb radiance measurements is described in Rozanov et al. (2024). In addition, atmospheric pressure profiles from the OMPS-LP data product of the Goddard Earth Sciences Data and Information Services Centre are used (Taha, 2020) to interpolate the aerosol extinction coefficient to specific pressure levels. These are forward



processing data from the NASA GSFC Global Modelling Assimilation Office (GMAO) at the nearest grid cell to each LP
event, which are interpolated at the corresponding measurement time.

SCIAMACHY and OMPS-LP have approximately 14 orbits per day and it takes 3–4 days to achieve full global coverage.
SCIAMACHY covers latitudes up to 60° N throughout the year. Latitudes north of 75° N are covered from the end of February
until October. The coverage extends to a maximum of 84.4° N. OMPS-LP covers latitudes above 60° N from the end of January
until November and latitudes north of 75° N from mid-March until October. The coverage extends to a maximum of 81.4° N.
For both data sets, retrieved profiles with a vertical resolution greater than 7 km are filtered out to ensure high-quality data at
all heights.

## 2.2  Particle size distribution parameters

Estimating the transported sulfate aerosol mass requires assumptions about the size distribution of the aerosols responsible
for the increase in extinction. Previous studies have sought to derive information regarding the size of particles using limb
scattering measurements (Rieger et al., 2014; Pohl et al., 2024). However, the retrievals are dependent on many assumptions
about scattering processes in the atmosphere and aerosol properties. Therefore, the stratospheric aerosol data set from SAGE
III/ISS was used, which has been used several times to determine information about the PSD (Wrana et al., 2021; Knepp et al.,
2024).

SAGE III on ISS is a solar occultation instrument (Cisewski et al., 2014) that was launched on 19 February 2017 and started
collecting data in June 2017. This instrument is an update of the SAGE III instrument on Meteor-3M (Thomason et al., 2010)
and is very similar in functionality. Version 6.0 of the SAGE III/ISS data set (NASA/LARC/SD/ASDC, 2025) is used for this
work, which contains PSD parameters derived according to the procedure described in Knepp et al. (2024).

In addition to the reduced spatial and temporal coverage, SAGE III/ISS observes high latitudes later in the year than the
Limb instruments. A common challenge faced by remote sensing instruments is the signal-to-noise ratio, which decreases with
altitude. The inversion method employed for SAGE III/ISS permits negative values of the extinction coefficient, resulting in an
increase in the proportion of negative coefficients with altitude. For SAGE III/ISS, at short wavelengths such as $\lambda = 449$ nm,
this results in a greater proportion of negative extinction coefficients at altitudes above approximately 32 km, relative to positive
extinction coefficients, thereby causing the mean extinction coefficient to become negative. This phenomenon has the potential
to introduce biases or make a retrieval in the mid-stratosphere infeasible.

## 2.3  ECMWF

The ECMWF ERA5 Reanalysis data is used to verify the atmospheric dynamics in specific periods (Hersbach et al., 2020).
ERA5 integrates model data and observations to create a uniform global data set. The data set "ERA5 hourly data on pressure
levels from 1940 to present" is used for this study. In this study, the parameters geopotential, potential vorticity, temperature,
specific humidity, ozone and northward and eastward wind are considered at a pressure level of 10 hPa, i.e. at an altitude of
approximately 30 km, during the polar vortex season.





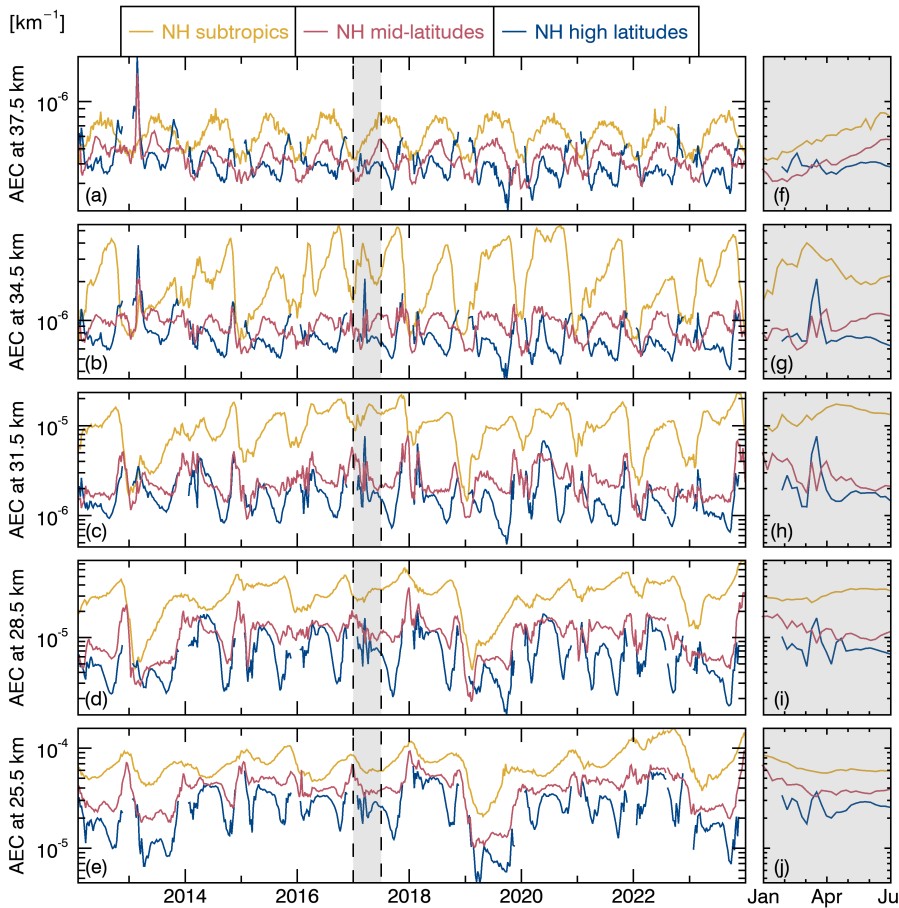

**Figure 1.** Weekly averaged aerosol extinction coefficient (AEC) at 869 nm from OMPS-UBr for five altitudes in the Northern Hemisphere (NH). The subtropics are defined from 20–40° N, the mid-latitudes from 40–60° N and the high latitudes from 60–80° N. The first half of 2017 is shaded within the dashed lines in (a–e) and zoomed in on the right-hand side in (f–j).

## 3 Results

### 3.1 Zonal mean aerosol extinction coefficient

Figure 1 shows the zonally averaged aerosol extinction coefficient (AEC) at 869 nm from OMPS-UBr in three distinct latitude ranges in the Northern Hemisphere at altitudes of 25.5, 28.5, 31.5, 34.5 and 37.5 km.

During the OMPS period, the tropical volcanic eruptions Kelut (2014), Ambae (2018), Soufriere St. Vincent (2021) and Hunga (2022) are expected to influence the aerosol extinction coefficient. Moreover, two eruptions in mid-latitudes had an impact on the stratosphere. These were the eruptions of Calbuco in the Southern Hemisphere in 2015 and of Raikoke in the Northern Hemisphere in 2019. The impact of volcanic eruptions in the stratosphere is closely associated with the altitude of the plume. Notably, the eruption of Hunga was the only one for which a significant aerosol plume was documented to reach





the mid-stratosphere. This plume could be observed up to an altitude of approximately 26 km (Duchamp et al., 2023). The impact of this eruption can be identified in Fig. 1 e. After the eruption, the aerosol extinction coefficient reaches its maximum at an altitude of 25.5 km in the northern subtropics during the OMPS period. The other eruptions referenced are not expected to exert a direct influence upon the mid-stratosphere.

In general, Fig. 1 shows a decrease in the aerosol extinction coefficient towards the pole. The meridional gradient and also
the minimum extinction coefficient at high latitudes are most pronounced in late summer and early autumn. In addition, the aerosol extinction coefficient in the middle and high latitudes shows a stronger variance in the polar vortex season compared to the summer months.

The occurrence of a local streamer event can be indicated by a jump in the weekly zonally averaged extinction coefficient at an altitude of 31.5 km, as observed in Fig. 1 c.

In the northern high latitudes, indicated by the blue line in Fig. 1, a sudden increase can be observed in the mid-stratosphere in the weekly zonally averaged extinction coefficient in the shaded area in the first half of 2017. The signal is strongest at an altitude of 31.5 km. At this altitude, the aerosol extinction coefficient exhibited a sixfold increase in spring 2017, marking the highest value recorded at this altitude within the northern high latitudes during the OMPS measurement period. At altitudes of 28.5 km and 34.5 km, the zonally averaged extinction coefficient increased by more than threefold. At an altitude of 28.5 km,
the aerosol extinction coefficient is higher in other years than during early 2017 due to a higher background aerosol level in those years. At an altitude of 34.5 km, the peak in the extinction coefficient in 2017 is only exceeded by the signature of the Chelyabinsk meteor (Gorkavyi et al., 2013; Rieger et al., 2014), which exploded in the mid-stratosphere four years earlier in the spring of 2013. This is also the only period during the OMPS mission in which aerosol extinction is greater at middle and high latitudes than in the subtropics. An increase in the aerosol extinction coefficient in spring 2017 can also be observed,
albeit to a lesser extent, at altitudes of 25.5 km and 37.5 km.

In the northern mid-latitudes, indicated by the red line in Fig. 1, two distinct yet less pronounced peaks can be identified in March and April 2017 at altitudes of 28.5 km, 31.5 km and 34.5 km. At an altitude of 31.5 km, a more than twofold increase in the aerosol extinction coefficient is observed in these two peaks. The two peaks can be observed before and after the strongest increase in the high latitudes.

**3.2 Aerosol extinction in the polar vortex season in early 2017**

In Fig. 2, the zonal mean eastward wind at 60° N and and the temperature from 65–90° N at 10 hPa for early 2017 are shown. The representation is based on Baldwin et al. (2021). Two minor warming events are recorded at the beginning and end of February. A sudden increase in temperature can be observed at high latitudes. In addition, there is a weakening of the wind, but no reversal of the wind direction at 60° N and 10 hPa.

Figure 3 shows the development of the weekly averaged extinction coefficient interpolated to 10 hPa, i.e. approximately 30 km altitude for the initial 16 weeks of 2017. The figure shows the aerosol extinction coefficient at 869 nm in bins of 5° latitude and 15° longitude in the Northern Hemisphere.





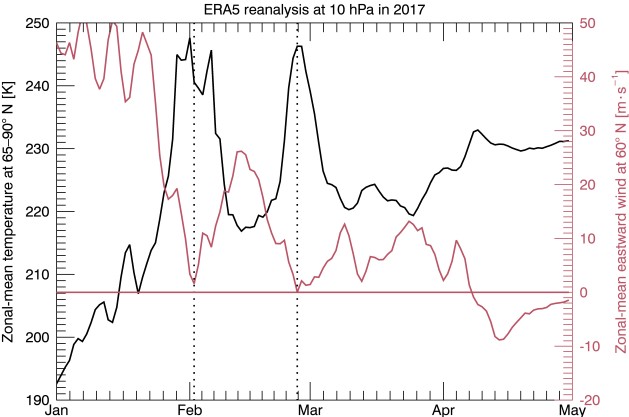

**Figure 2.** SSW criteria for spring 2017. The zonal mean eastward wind at 60° N is represented in red and the mean temperature from 65–90° N at 10 hPa is shown in black. Two minor SSWs in February are marked by dotted lines.

At the beginning of 2017, the aerosol extinction coefficient in the mid-latitudes is subject to considerable variation. Figures 3 b and 3 e show tongue-like patterns of increased aerosol extinction coefficient, extending from the subtropics to the mid-
latitudes from west to central Asia. The increase in aerosol extinction that occurred subsequent to the second of these tongues in early February coincides with the first minor warming in 2017 (see Fig. 2), which caused a weakening and deflection of the vortex. At this time, the centre of the polar vortex was located at approximately 70° N. The increase in the aerosol extinction coefficient cannot be traced further in the following weeks.

On 25 February, a late minor SSW occurred (see Fig. 2). At a pressure level of 10 hPa, this had the effect of a temperature
increase at the North Pole of more than 30 K. This phenomenon resulted in a weakening and separation of the polar vortex. A shift of the vortex edge into the subtropics can be observed from around 1 March. The vortex was subsequently elongated and displaced off the pole.

For a better understanding of the dynamic of the polar vortex, Fig. 4 presents in the lower five rows ERA5 reanalysis data for a four-week period in March 2017 following the second minor warming. The parameters under consideration are the wind,
the potential vorticity with contour levels of geopotential height, the specific humidity, the ozone mass mixing ratio and the temperature, all at 10 hPa. The vortex centre itself is associated with low wind speed, geopotential height and ozone mixing ratio, and high potential vorticity and specific humidity. After the two minor SSWs, the polar vortex is no longer associated with low temperatures.

Looking at the first column of Fig. 4 for the week of 5–11 March, an increase in wind speed towards the pole of the now
very uneven vortex can be observed. Following the displacement of the polar vortex towards the subtropics, the transportation of aerosols from the subtropics towards the pole can be deduced from the increased extinction coefficient at the edge of the vortex over Asia, see Figs. 4 a and 4 e. Subtropical air is transported along the edge of the polar vortex, and an area of high geopotential height or relative high pressure is formed, see Fig. 4 i. An anticyclone forms around the area of high geopotential



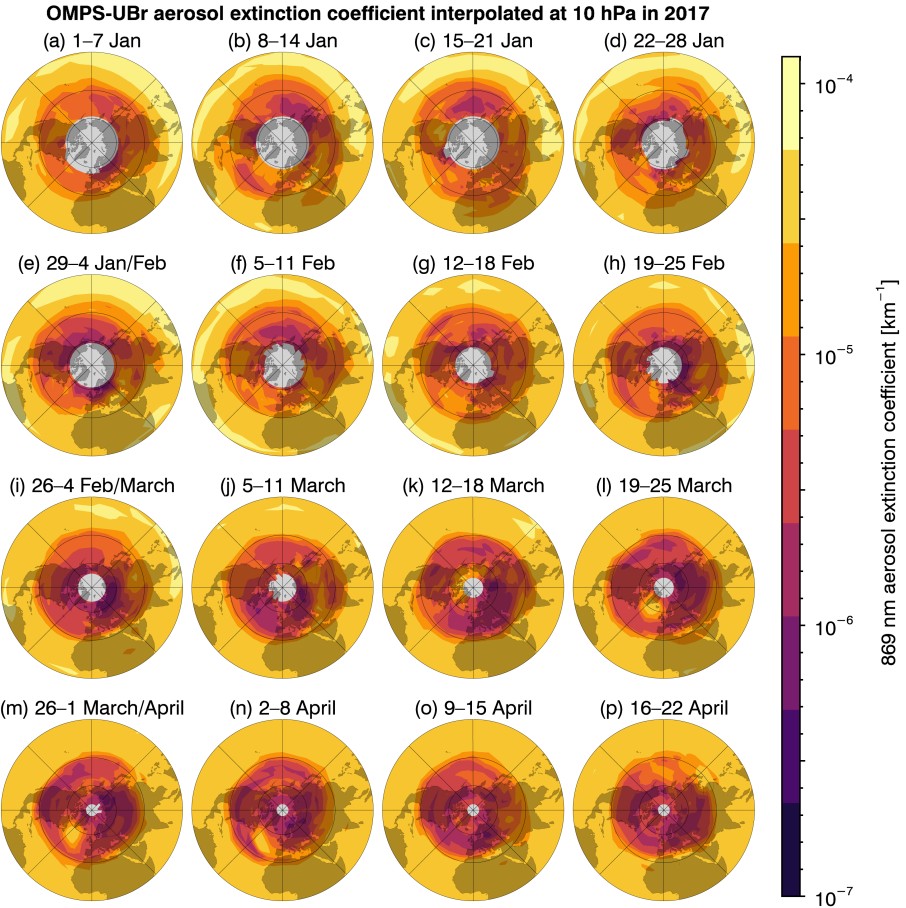

**Figure 3.** Polar stereographic maps of the weekly averaged aerosol extinction coefficient at 869 nm from OMPS-UBr interpolated to 10 hPa in early 2017 in bins of 5° latitude and 15° longitude for the entire Northern Hemisphere.

height. Areas of an increased aerosol extinction coefficient correlate with a low potential vorticity and the air parcel is dry and
initially ozone-rich.

As of 16 March, the material is observed to depart from the edge of the polar vortex. Concurrently, an anticyclone is established over Greenland (70° N, 45° W), see Figs. 4 g and 4 k. This structure persists and maintaining the material's position for nearly three weeks. Subsequently, from approximately 6 April onwards, it can still be observed over the North Atlantic (45° N, 45° W). It is notable that the air parcel remains dry for the entire time that the signal is observable in the aerosol
extinction coefficient. In the ozone mass mixing at this pressure level, the air parcel is no longer richer in ozone than the ambient air after a period of one week, at which point rapid mixing occurs. There is no evident correlation between increased aerosol extinction and temperature.



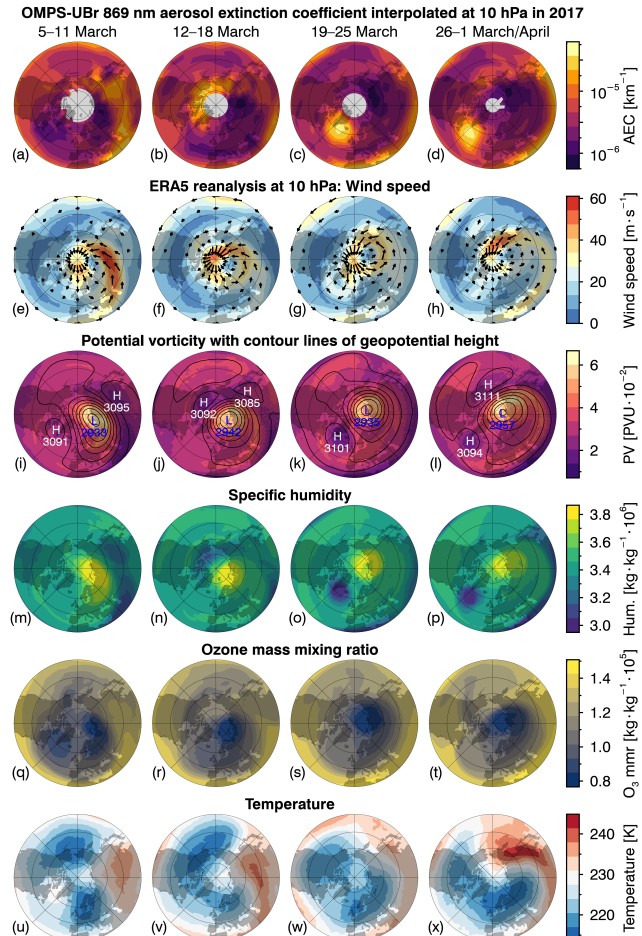

**Figure 4.** Polar stereographic maps from 30–90° N, with 0° E as the central longitude line from bottom to centre, for various weekly averaged parameters (rows) at 10 hPa for a period of 4 weeks in March and April 2017 (columns). First row contains the aerosol extinction coefficient at 869 nm from OMPS-UBr as in Fig. 3. Rows 2 to 6 show maps of ERA5 reanalysis data, including wind speed (e–h), potential vorticity in potential vorticity units and geopotential height in decametres (dam) (i–l), specific humidity (m–p), ozone mass mixing ratio (q–t) and temperature (u–x). Wind arrows are plotted at wind speeds of 10 m s$^{-1}$ or higher. The contour lines of the geopotential height are delineated at intervals of 0.2 km. The highest and lowest geopotential heights are denoted with 'H' and 'L', respectively, along with the corresponding height in dam.

After 19 March, the formation of another streamer can be observed in a substantial area of low PV, low specific humidity and high ozone mass mixing ratio at 30° N over Asia. After 26 March, Figs. 4 h, 4 l and 4 p show the transport of this subtropical

air to mid-latitudes. As can be seen in 4 l, a further area of high geopotential height has formed over the Bering Sea.

Figure 5 shows a comparison of the vertical profiles of the aerosol extinction coefficient at middle to high latitudes, both inside and outside the anticyclone, and in the tropics over central and northern Africa at the beginning of March before the start



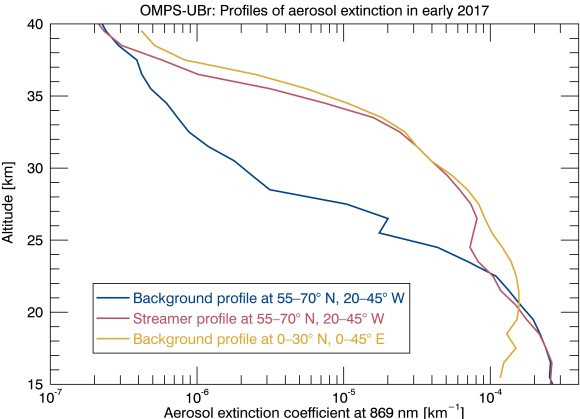

**Figure 5.** Weekly averaged profiles of the aerosol extinction coefficient in middle to high latitude (55–70° N, 20–45° W) for background conditions before the streamer event (05–11 March) and inside the anticyclone during the streamer event (19–25 March), and in the tropics to subtropics (0–30° N, 0–45° E) before the start of the streamer event (26 February to 04 March).

of the streamer event. To make this comparison, profiles from the same geographical region (55–70° N and 20–45° W) were averaged for different weeks before and during the streamer event, as in Figs. 3 j and 3 l. To compare these with the vertical

profile of the aerosol extinction coefficient from the region in which the streamer likely originated, measurements before the start of the streamer event (see Fig. 3 i) from the tropics to subtropics (0–30° N, 0–45° E) were averaged.

The edge of the tropical air parcel which is transported within a high-pressure area towards high latitudes creates a strong transport barrier between the anticyclone and the ambient air. At an altitude of 24–38 km the vertical profiles in the tropics and in the anticyclone differ only slightly. This finding suggests that the mixing process is minimal for the streamer. This transport

barrier is also the reason why the increase in aerosol extinction can be observed for so long. The agreement between anticyclone and subtropics is greatest at 30.5 km, the altitude at which the strongest signal from the streamer, i.e. the biggest percentage increase in the aerosol extinction coefficient, is visible. Up to an altitude of around 26 km, there is a clear distinction in the profiles of subtropical and high-latitude regions. Conversely, a high degree of similarity is observed between the profiles both within and outside the anticyclone up to this altitude. At an altitude of 32.5 km, the aerosol extinction coefficient increases by

a factor of 27 from the background level to the streamer.

In addition to 2017, anticyclones in the mid-latitudes are also evident in the aerosol extinction coefficient in other years, e.g. 2015 and 2022, albeit to a lesser extent.

As of 13 March, a weak polar vortex streamer becomes discernible in the reanalysis data through an increased potential vorticity and specific humidity below 45° N over Southern Europe, see Figs. 4 j and 4 n. This is not evident in the aerosol

extinction coefficient.



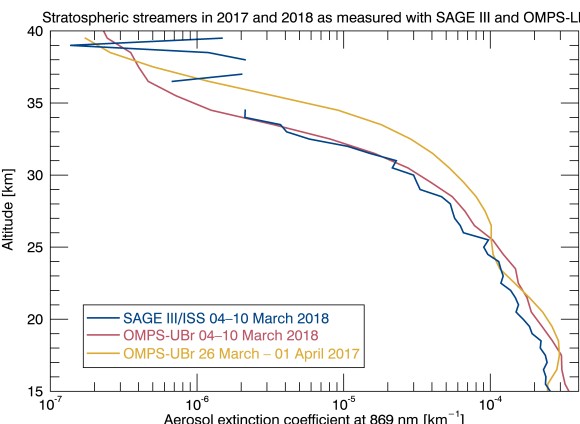

**Figure 6.** Mean vertical profiles of aerosol extinction at 869 nm for two streamers in mid-latitudes from 26 March to 01 April 2017 (45–55° N, 25–50° W) and from 04–10 March 2018 (45–55° N, 10° W to 15° E) as retrieved from OMPS-LP and SAGE III/ISS measurements.

### 3.3 Estimation of the transported aerosol mass in 2017

For the estimation of the transported aerosol mass, it is important to note that measurements with SAGE III on ISS began in June 2017, precluding the possibility of direct overlap with the observed signatures in 2017 and earlier. Consequently, the PSD parameters for comparable atmospheric conditions in subsequent years must be used. To this end, a streamer event from 2018 was selected. It began in February 2018, a few weeks earlier in the season than in 2017, and was observable with SAGE III/ISS in early March. Below 31 km altitude, the streamer had a comparable zonal influence on the aerosol extinction coefficient as in 2017 and was also briefly located in the high latitudes, see Fig. 1. Since there was no volcanic eruption between the two events in 2017 and 2018 that could have affected the stratosphere, the background aerosol level remained consistent. The present study focuses on the streamer event in early 2017 due to its greater vertical extent, its stronger influence on high latitudes, and the fact that there was negligible mixing taking place due to the anticyclone.

Figure 6 shows vertical profiles of the two streamers in 2017 and 2018 in the mid-latitudes (45–55° N) for different longitude regions as measured with OMPS-LP and SAGE III/ISS. Both streamer profiles originate from a time approximately 3–4 weeks after the start of the streamer event. Up to an altitude of 34 km, there is very good agreement between the retrieved aerosol extinction coefficient with OMPS-UBr and SAGE III/ISS in 2018. This indicates that the streamer event in 2018 can be observed with both instruments. The vertical profile from 2017 shows a higher average aerosol extinction over the entire altitude range of the streamer. This can be partially explained by the fact that the streamer in 2017 is more compressed within an anticyclone, thereby concentrating the aerosols in a smaller area.

Nevertheless, this period in March 2018 and this region appears to be the best possible match with the streamer event in 2017 in the measurement period of SAGE III. For this selection, the mean median radius and geometric standard deviation of a monomodal log-normal distribution are obtained for every altitude step. The Mie code of the University of Oxford is used to calculate the extinction cross section for these PSD parameters.





The time period spanning 14 days from 19 March to 1 April is selected for the estimation of the transported sulfate aerosol mass in spring 2017. This is due to the aerosol layer exhibiting minimal zonal or meridional movement during this interval, thereby facilitating its observation over the North Atlantic. The mean extinction coefficient at 869 nm is calculated for each

height level from 24.5–38.5 km for bins of size 3° × 3°. Furthermore, the mean extinction coefficient from 0–270° E, i.e. in the range in which no increase in the aerosol extinction coefficient due to transported material can be observed, is calculated for each latitude step. Bins are designated as an anomaly if the extinction coefficient is higher than its latitudinal mean plus three times the standard deviation and if an increase is also observed in the surrounding bins.

For each such anomaly, the absolute number of particles $N(z)$ at height $z$ is calculated. The height $z$ is defined as the

centre height of an altitude layer spanning 1 km. The volume $V$ of the respective bin and the number density $N_0(z)$ are used to calculate the absolute number of particles $N(z)$. The number density is derived from the additional aerosol extinction attributable to the streamer and the PSD parameters. The mass of a log-normally distributed aerosol population can then be calculated using the 3rd moment of the log-normal distribution (Grainger, 2023).

$$m(z) = \frac{4}{3}\pi \cdot N(z) \cdot r_{\mathrm{med}}^3 \cdot \rho(z,T) \cdot \exp\left(\frac{9}{2}\ln(\sigma_g)^2\right) \tag{1}$$

In this equation, $m(z)$ denotes the total mass of the aerosol population at height $z$ and $\rho(T)$ represents the density of the aerosol particles as a function of temperature $T$ at height $z$. For the mass density $\rho$, the measured mass densities of sulfuric acid for different mass fractions and temperatures of Myhre et al. (1998) are interpolated for an $H_2SO_4$ mass fraction of 75 % and for the respective GMAO temperature at this altitude.

The aerosol mass that has been transported in 2017, as calculated with averaged PSD parameters for measurements of the

streamer event in 2018, is estimated to be approximately 0.99 kt in the altitude range of 24.5–38.5 km, where the aerosol extinction signal of this air parcel is observable.

Due to the strong mixing barrier, it can be assumed that the mass changes are negligible along the transport path. For the high latitudes defined as 60–90° N, the average PSD parameters from SAGE III/ISS for the same altitude range result in a total mass of 1.34 kt for background conditions. This means that the streamer event in the week of 12–18 March causes a zonal

increase in the total aerosol mass in the altitude range of 24.5–38.5 km by about 74 %. At an altitude of 30.5 km, the aerosol mass in the high latitudes zonally increases by about 260 % during the same period. On a local scale, the mass can increase by more than one order of magnitude.

### 3.4 Aerosol extinction in the polar vortex season in early 2005

A notable peak in the aerosol extinction coefficient in the mid-stratosphere during the SCIAMACHY measurement period was

observed in spring 2005. Figure 7 shows the aerosol extinction coefficient at 750 nm in the Northern Hemisphere interpolated at an altitude of 30.5 km from spring to late-summer 2005. The northern winter of 2004/2005 was relatively cold with prolonged periods of temperatures below 200 K in the lower stratosphere and characterized by a strong and stable polar vortex (Sonkaew et al., 2013). On 13 March, there was an exceptionally early final warming, approximately one month before the





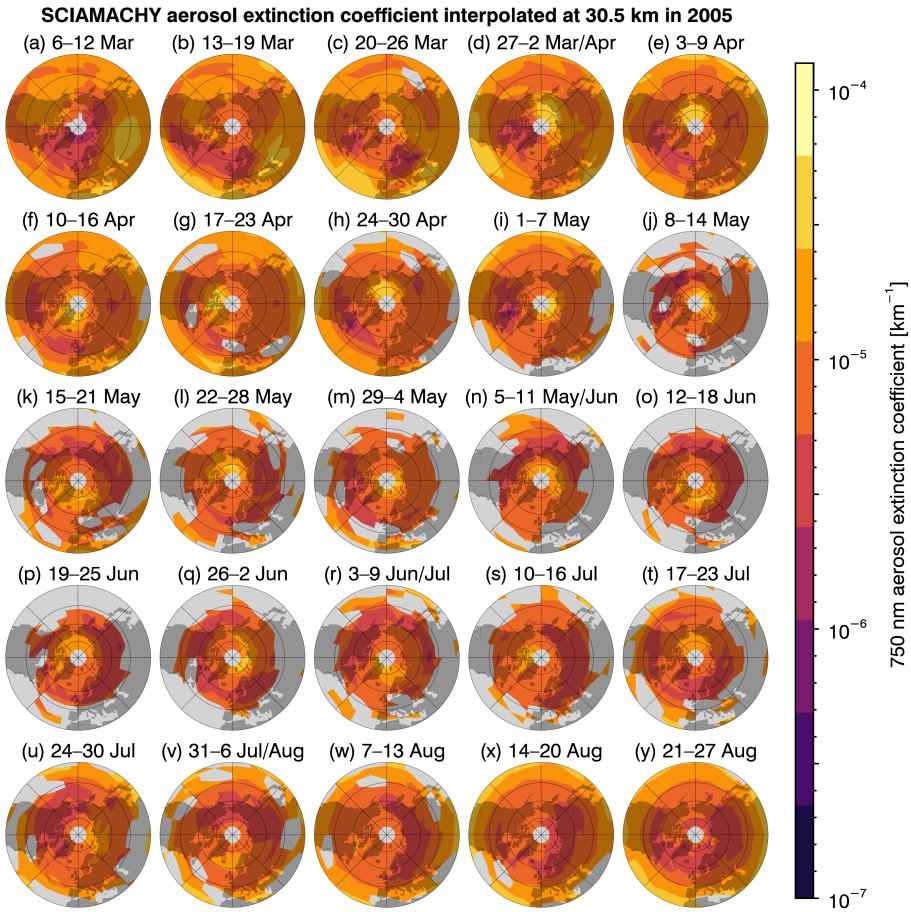

**Figure 7.** Polar stereographic maps of the weekly averaged aerosol extinction coefficient at 750 nm from SCIAMACHY interpolated at 30.5 km in from March to August 2005 in bins of 5° latitude and 15° longitude from 30–90° N.

historical median. As can be seen in Figs. 7 a–d, there are some variations in March 2005 in the mid-latitudes. From the end of
March/beginning of April an increase in the aerosol extinction coefficient at the pole is noticeable.

Figure 8 presents in the lower five rows reanalysis data for the four weeks from mid-March to early April following the final warming. Before the final warming, a slightly increased aerosol extinction coefficient can already be observed in the mid-latitudes across a substantial geographical area extending from North America to the Pacific region, see Fig. 8 a. This suggests that tropical air has already accumulated in a high-pressure area extending over a considerable region, see Fig. 8 i.
In the weeks following the final warming, the region of low geopotential height of the collapsed polar vortex shifts towards mid-latitudes, see Fig. 8 j. Concurrently, the low-pressure area is compressed and shifts towards the pole until the end of March. By then, a strengthening anticyclone develops with low PV (Fig. 8 l), dry ozone-rich air (Figs. 8 p and 8 t) and high aerosol extinction (Fig. 8 d) over the Arctic Ocean near Siberia. The anticyclone shifts towards the pole and remains there trapped in the circulation for a period of several months until August of the same year, see Fig. 7.





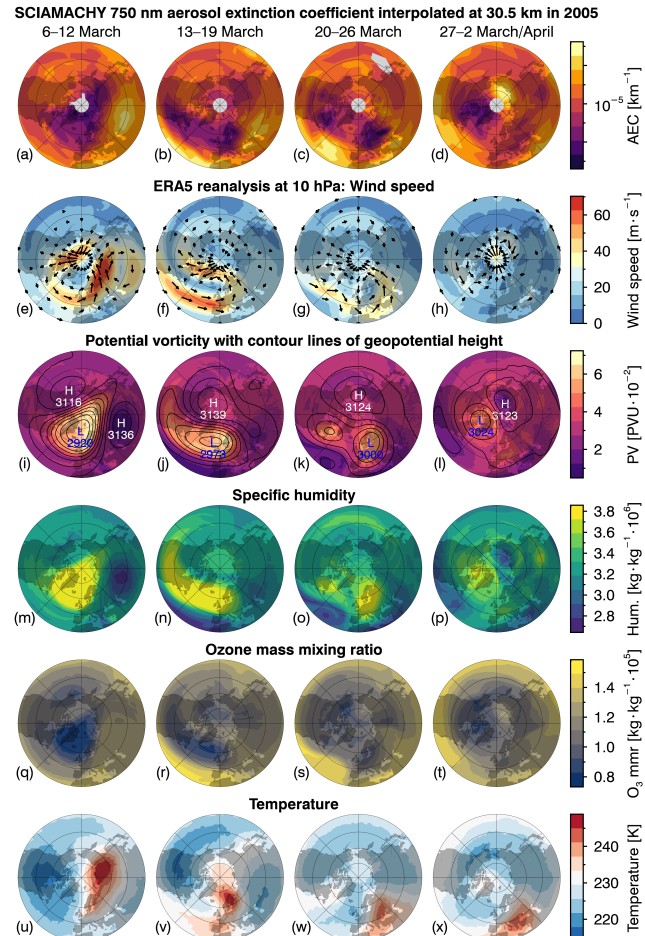

**Figure 8.** Polar stereographic maps from 30–90° N for various weekly averaged parameters (rows) for a period of 4 weeks in March and April 2005 (columns). First row contains the aerosol extinction coefficient at 750 nm from SCIAMACHY as in Fig. 7 at an altitude of 30.5 km. Rows 2 to 6 show maps of ERA5 reanalysis data at 10 hPa, including wind speed (e–h), potential vorticity in potential vorticity units and geopotential height in dam (i–l), specific humidity (m–p), ozone mass mixing ratio (q–t) and temperature (u–x). Wind arrows are plotted at wind speeds of 10 m s$^{-1}$ or higher. The contour lines of the geopotential height are delineated at intervals of 0.2 km. The highest and lowest geopotential heights are denoted with 'H' and 'L', respectively, along with the corresponding height in dam.

Similar structures to those seen in 2005 can be found in other years in the aerosol data sets examined, e.g. in 2006, 2007 and 2014.



## 4   Discussion

Both described events in 2005 and 2017 can be explained by the combination of streamers, transporting tropical air into the middle or high latitudes and anticyclones holding material in these regions.

The exact extent and shape of the polar vortex and how far it reaches into the mid-latitudes appears to be decisive for the transport of aerosols towards the pole. In the majority of years in the observation period 2003–2023, a transport of tropical air into the mid-latitudes and a variation in the aerosol extinction especially in mid-latitudes can be observed. The intensity depends largely on the background level in tropical aerosol extinction. The duration and stability of a streamer depend on the dynamics in mid-latitudes in connection with the polar vortex and the formation and strength of the anticyclone, which prevents mixing with the surrounding air.

During early 2017, two minor warmings occurred in February 2017. The impact of the former on the mesosphere was described in Eswaraiah et al. (2020). A streamer event can be observed in the stratospheric aerosol extinction coefficient following this first warming at the beginning of February, during which tropical air is transported to mid-latitudes. However, this signal is only observable for approximately one week, and the high latitudes are not adequately covered by OMPS-LP at the beginning of February. Consequently, the estimation of transport was only possible via surrounding latitudes on the outward or return transport from the pole. The reanalysis data indicates that no anticyclone is formed, thereby ensuring that the transported air parcel of the streamer does not remain concentrated at middle and high latitudes. Moreover, the signal in the specific humidity is no longer clearly observable after one week.

A notable characteristic of the polar vortex season 2017 is the comparatively late occurrence of a warming, i.e. the second minor warming, which was not also the final warming in the vortex season (Gogoi et al., 2023). This phenomenon enables the observation of the signal of an increased extinction coefficient above 75° N. The sequence of weakening, deflection and brief but repeated strengthening of the elongated vortex with high wind speeds in the direction of the pole and low speeds away from the pole ensures that material can accumulate at the pole. The vortex weakens once more, causing the material to remain in the mid-latitudes for several weeks. The formation of a high-pressure system, accompanied by the establishment of an anticyclone, results in the accumulation of material south of Greenland over the Atlantic.

Following the second warming, no correlation of low temperatures with the vortex or high temperatures with the streamer is discernible in the temperature profile. This exemplifies the significant weakening of the polar vortex, which did not regain its closed circular structure and strength following the two minor SSWs. In addition, the temperature gradient between the tropics and high latitudes is only marginal, which is why the transported air mass is no longer associated with increased temperatures.

The occurrence of tropical-subtropical streamers is not exclusive to a specific period during the vortex season. However, as the year progresses, the Limb instrument's coverage of the high latitudes diminishes, thereby impeding the comprehensive tracking of a structure's trajectory.

It is important to note that, in contrast to the more prevalent occurrence of streamer events in the Northern Hemisphere, these events are less frequent in the Southern Hemisphere due to the presence of a more stable vortex. The more-frequent observation of this phenomenon in the Northern Hemisphere may also be explained, by the elevated background level of the





aerosol extinction coefficient in the mid-stratosphere in the southern mid-to-high latitudes. This results in a lower gradient of the aerosol extinction coefficient between the South pole and the tropics compared to the gradient between the North pole and the tropics. Consequently, the presence of streamer events in the Southern Hemisphere is less pronounced in the aerosol extinction coefficient. However, this observed difference in the gradient may also result from increased retrieval uncertainties

in the Southern Hemisphere (Chen et al., 2020). The scattering angles of aerosols are strongly dependent on latitude for OMPS-LP measurements and change during the orbit. The observed signal in the Southern Hemisphere originates, among other things, from aerosol scattering in the sideways and backward directions, while in the Northern Hemisphere, forward scattering predominates.

    The observed signals show good agreement with the characteristics expected from model runs for stratospheric streamers

(Krüger et al., 2005). These also showed a higher probability of streamers occurring when the polar vortex shifted equatorward. The dimensions of the streamer in 2017, with a length of over 10,000 km in the development phase, a width of about 1,500 km and a vertical extent of about 14 km, correspond very well with the modelled extent of streamers. The duration of the event, spanning five weeks, is at the upper end of the expected time frame. This can be attributed to the anticyclone in 2017, which kept the tropical air in mid-latitudes for an extended period of time. The observation made in 2017 of a transport to 70–75° N

also occurs in the model in some years. In contrast to the model, no evidence of irreversible mixing at high latitudes is observed in 2017. The streamer in 2017 is located over East Asia, where the model predicts the maximum number of streamer events.

    It is possible to observe different tracers of streamers and anticyclones for varying lengths of time. Ozone mixing, for instance, is a highly potent process that occurs in March 2017 a few days after the start of the streamer event. This mixing occurs more rapidly at higher altitudes (10 hPa) than at lower altitudes (20–30 hPa). At these lower altitudes, the air mass is

still rich in $O_3$ even at the end of March. This observation is consistent with Thiéblemont et al. (2011), who also demonstrated in their description of a FrIAC in 2007 that the ozone mixing ratio decreases much faster at 10 hPa than at 20 hPa. The ozone signal is more strongly linked to gas chemistry and photochemistry than to mixing processes. The process of relaxation to lower ozone levels is even faster in spring, due to higher levels of solar radiation (Manney et al., 2006). Conversely, elevated aerosol levels and diminished levels of specific humidity persist within the anticyclone for a period of approximately one month. The

correlation of anticyclone and dry air in the specific humidity is observable up to higher altitudes (5–20 hPa).

    During the winter of 2016/17, the quasi-biennial oscillation (QBO) phase at 10 hPa shifted from westerly phase of the QBO (WQBO) to the easterly phase of QBO (EQBO). This means it was effectively neutral during the season. In general, signals from FrIACs (Thiéblemont et al., 2011) or streamers are more pronounced in years with an EQBO phase. This is due to a higher background level of aerosol extinction in the tropical mid-stratosphere in EQBO years, creating a greater gradient in

the extinction coefficient between the tropics and the North Pole. The polar vortex tends to be stronger during the WQBO phase while being weaker and more unstable during the EQBO phase (Gray et al., 2018). Upward-propagating Rossby waves have a stronger orientation towards the polar vortex during the EQBO phase. A stronger interaction with the mean flow is associated with a higher probability of sudden stratospheric warming events (Baldwin et al., 2021). However, the QBO phase is not a definitive criterion, as Thiéblemont et al. (2016) show based on an observed FrIAC in 2013, while the QBO phase was

westerly. The strong streamer event observed in 2017 during a neutral QBO phase also suggests that the QBO phase is not





decisive for the occurrence and strength of streamer events. This is consistent with the observation in Waugh (1996) that the QBO phase has only a minor effect on winds poleward of 20° and thus on quasi-horizontal transport from the tropics.

The FrIAC in 2005 has already been described in detail based on observations (Manney et al., 2006) and modelling (Allen et al., 2011). As in Manney et al. (2006) for the trace gases $N_2O$ and $H_2O$, the signature of the FrIAC can be observed in the

extinction coefficient until August.

Similar to 2005, FrIACs of varying intensity can also be seen in the Northern Hemisphere in 2006, 2007 and 2014 (not shown), subsequent to the breakdown of the polar vortex. These circumpolar areas can be identified by a slight increase in the extinction coefficient at high latitudes.

## 5   Sources of error

It is conceivable that the observed signal is an artefact of retrieval or a specific measurement technique. The transport in 2017 can be observed in a very similar way in the other OMPS-LP retrieval products from NASA and the University of Saskatchewan. The FrIAC in 2005 can also be observed using aerosol extinction data collected by SCIAMACHY. This suggests that the observed phenomena are not artefacts of a single retrieval or instrument. However, both phenomena are detected using a similar measurement technique. For comparison with the aerosol extinction coefficient from a solar occultation instrument, a

streamer event from 2018 was selected, which could be observed by both SAGE III/ISS and OMPS-LP, see Fig. 6.

Up to an altitude of 34 km, there is very good agreement between the two profiles of the different satellite instruments. The observation of increased aerosol extinction profiles up into the mid-stratosphere at mid-latitudes using two instruments with different measurement techniques and retrieval methods, indicates that this is a genuine aerosol extinction signal.

The likelihood of a error in retrieval due the effects of water vapour and ozone during the streamer event is negligible, as

the two trace gases exhibit only very low absorption at the wavelength channels used. Increased ozone levels at an altitude of 30 km and above can only be observed during the first week of the streamer's presence. Additionally, the streamer is drier than the surrounding air.

The estimated amount of just below 1 kt depends heavily on the number density of the aerosols, which in turn depends on the assumptions made about the size distribution, i.e. the median radius and geometric standard deviation. For this estimation,

the "weighted median PSD parameters" from the SAGE III/ISS dataset (Knepp et al., 2024) were employed for every altitude step when a streamer in 2018 was measured in mid-latitudes with SAGE III/ISS.

This is a very rough estimate. Therefore, the assumptions about the PSD parameters were varied. To vary the PSD parameters, a single averaged PSD combination from the altitude range of 25–30 km was applied to the entire altitude range. This is owing to the fact that PSD retrieval is more reliable at lower altitudes than at high altitudes. In addition, PSD parameters from 0–30° N

from the entire SAGE III/ISS measurement period up to the Hunga eruption were averaged for each altitude step. However, both methods result only in a negligible deviation of about 5 % in aerosol mass.

Nevertheless, the estimate remains uncertain. No solar occultation measurements are available for spring 2017. The true PSD of aerosols above 30 km altitude is largely unknown and is subject to high bias and error when retrieved from SAGE



measurements. In addition, there is general uncertainty about the true shape of the size distribution, which is assumed to be
monomodal for the retrieval. However, if the true shape is different, e.g. bimodal, this can lead to significant deviations in the
retrieval of PSD parameters (von Savigny and Hoffmann, 2020).

## 6 Conclusions

Stratospheric tropical-subtropical streamers are horizontal transport processes by which atmospheric trace substances can be
transported over long distances. Streamers are associated with disturbances in the polar vortex and an elongation of vortex
towards the subtropics.

Such streamers transport tropical air masses to middle and high latitudes, enabling them to cross the subtropical transport
barrier. The transport to mid-latitudes is not an uncommon phenomenon and occurs to a small extent multiple times every
year. However, the observation of a tropical-subtropical streamer reaching high latitudes is a rare occurrence. As shown for the
first time in this study, the aerosol extinction coefficient also exhibits a high correlation with streamer events. The association
of streamers with a low potential vorticity, high geopotential height, low specific humidity and high ozone mixing ratio has
already been demonstrated in previous studies.

Satellite instruments such as OMPS only cover high latitudes at the beginning and end of winter. Therefore, the time window
during which the transport of aerosols towards the pole can be tracked in detail is limited. The transport of increased aerosol
concentrations in the mid-stratosphere to high latitudes is not expected to significantly impact radiative transfer.

Anticyclones can hold low latitude air in middle or high latitudes for periods ranging from weeks to several months. This
phenomenon may occur both during the polar vortex season, such as in 2017, or after the polar vortex breakdown, as was the
case in 2005.

The most significant poleward aerosol transport by streamers observed in the last twenty years occurred in March 2017,
when about 1 kt of sulfuric acid aerosols were transported towards the middle and high latitudes. This almost doubled the
aerosol mass in high latitudes for the altitude range of 24.5–38.5 km of the streamer. On a local scale, the aerosol extinction
coefficient increased by a factor of 27.

*Data availability.* The Level 2 aerosol extinction coefficient data products of SCIAMACHY V3.0 and OMPS-LP V2.1 from the Institute of
Environmental Physics at the University of Bremen are available at https://www.iup.uni-bremen.de/DataRequest (Rozanov, 2024, login re-
quired). The SAGE III/ISS data used within this study are available on NASA's Atmospheric Science Data Center (NASA/LARC/SD/ASDC,
480 2025).

*Author contributions.* CL handled the different datasets and wrote the initial text for the paper. CL, RE and CvS discussed the observations
and interpreted the results. AR and CP provided the SCIAMACHY and OMPS-LP data. All authors contributed to the scientific discussion
and improvement of the initial paper.



*Competing interests.* The authors declare that they have no conflict of interest.

*Acknowledgements.* This work was funded by the Deutsche Forschungsgemeinschaft (research grant O-MSP-LI, project no. 516357253). We want to acknowledge support by the University of Greifswald and thank the Earth Observation Data Group at the University of Oxford for providing the IDL Mie routines used in this study. The University of Bremen team was funded in part by the European Space Agency (ESA) via the CREST project, by the German Research Foundation (DFG) via the research unit VolImpact (grant no. FOR2820), and by the University of Bremen and state of Bremen. The authors gratefully acknowledge the computing time granted by the Resource Allocation

Board and provided for the supercomputers Lise and Emmy at NHR@ZIB and NHR@Göttingen as part of the NHR infrastructure. The calculations for this research were conducted using computing resources under project no. hbk00098.



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
