# Peer review of "Observations of stratospheric streamers and frozen-in anticyclones in aerosol extinction"

_EGUsphere, 2025_

## Author Comment (AC1)

**Reply to a comment by Lynn Harvey**

The authors should cite this study in the background section: Harvey, V. L., M. H. Hitchman, R. B. Pierce, T. D. Fairlie (1999). Tropical aerosol in the Aleutian High, JGR, <a href="https://doi.org/10.1029/1998JD200094">https://doi.org/10.1029/1998JD200094</a>

Thank you for pointing that out. We had overlooked the paper and it definitely needs to be included, as it contains information on both poleward transport from the subtropics and observability in aerosol extinction. We have now added a reference to Harvey et al. 1999 in the introduction:

"The poleward transport of tropical aerosols at an altitude of 15–30 km was already described in Harvey et al. 1999. Here, SAM II measurements of the aerosol extinction coefficient at 1000 nm revealed a reduced surface aerosol density (SAD) in the polar vortex during the winter months from 1978 to 1991, while the SAD in the Aleutian High was increased."

---

## Author Comment (AC2)

**Replies to comments by Referee 1 Daniele Visioni**

Comments, replies, "changes in the manuscript"

References to added publications not already cited in the first version of the paper are listed in the replies.

In this paper, Löns et al use anomalies in stratospheric aerosol extinction, detected using multiple satellite products, to track the transport of air towards the poles, in particular through streamers – persistent ozone-rich masses of air coming from sub-tropical latitudes – and in particular look at the behavior of one identifiable event using this methodology in 2017. I found this paper extremely interesting and well written, and the analyses robust, and therefore recommend publication after some comments I attach below are addressed.

Thank you very much for your clear and helpful comments.

L. 9: "in that year" rather than "this year" (might be confusing)

Done.

L. 43: define what the "tape recorder" is, a reader might not necessarily be familiar with this term.

In combination with the community review to shorten the introduction, the reference to the tape recorder signature has been omitted.

L. 185: this section title shouldn't be "ECMWF", that's just the name of the Center.

Title changed to "ERA5 reanalysis data"

L. 187: the dataset needs to be cited following Copernicus, not just with the long title, and the reference (with its own DOI) needs to be added to the Data Availability section.

Both done.

Related to ERA5, it would be useful to include some references that validate ERA5 stratospheric transport. Here are a few suggestions, which the authors should try to include and talk about in Section 2.3:

Vogel, B., Volk, C. M., Wintel, J., Lauther, V., Clemens, J., Grooß, J.-U., Günther, G., Hoffmann, L., Laube, J. C., Müller, R., Ploeger, F., and Stroh, F.: Evaluation of vertical transport in ERA5 and ERA-Interim reanalysis using high-altitude aircraft measurements in the Asian summer monsoon 2017, Atmos. Chem. Phys., 24, 317–343, https://doi.org/10.5194/acp-24-317-2024, 2024.

Ploeger, F., Diallo, M., Charlesworth, E., Konopka, P., Legras, B., Laube, J. C., Grooß, J.-U., Günther, G., Engel, A., and Riese, M.: The stratospheric Brewer–Dobson circulation inferred from age of air in the ERA5 reanalysis, Atmos. Chem. Phys., 21, 8393–8412, https://doi.org/10.5194/acp-21-8393-2021, 2021.

Xiaozhen Xiong, Xu Liu, Wan Wu, K. Emma Knowland, Qiguang Yang, Jason Welsh, Daniel K. Zhou, Satellite observation of stratospheric intrusions and ozone transport using CrIS on

**SNPP, Atmospheric Environment, Volume 273, 2022, 118956, ISSN 1352-2310, <a href="https://doi.org/10.1016/j.atmosenv.2022.118956">https://doi.org/10.1016/j.atmosenv.2022.118956</a>.**

Thank you for all your remarks on this subsection. The paragraph has now been expanded and the publications of Ploeger et al. 2021, Diallo et al. 2021 and Vogel et al. 2024 on the validity of the ERA5 dataset for stratospheric transport have been included:

"ERA5 data products are widely used and generally provide a good representation of transport processes in the atmosphere. Several studies have validated stratospheric transport in ERA5. Age of air analyses were used to investigate the representation of the Brewer-Dobson circulation in ERA5 and to compare it with observational data (Ploeger et al. 2021). Here, in the lower and middle stratosphere in the Northern Hemisphere, there are indications that ERA5 is at the upper edge of the observational uncertainty in terms of mean age of air and that the Brewer-Dobson Circulation therefore appears to be low biased, also in comparison to the previous version ERA-Interim (Diallo et al. 2021). ERA5 was also used to verify the upward transport of greenhouse gases and pollution during the Asian monsoons via age of air and trajectory analyses (Vogel et al. 2024). Comparison with in situ data showed that the transport into the lower stratosphere was well represented."

**References:**

Diallo, M., Ern, M., and Ploeger, F.: The advective Brewer–Dobson circulation in the ERA5 reanalysis: climatology, variability, and trends, Atmospheric Chemistry and Physics, 21, 7515–7544, <a href="https://doi.org/10.5194/acp-21-7515-2021">https://doi.org/10.5194/acp-21-7515-2021</a>, 2021.

L. 295: it would be useful to add some more details (and a reference here to a volcanic dataset) where a reader can see the location and magnitude of the mentioned eruptions. I also think the claim that only Hunga reached the mid-stratosphere is a bit stretched, and the authors could provide some more details: in Li et al. (2023) for instance, the aerosols from La Sourfrier are clearly shown to reach 22-23 km.

This section was insufficiently referenced and the significance of our emphasis was unclear. The list of volcanic eruptions was meant to show that the direct volcanic influence is expected to be negligible in the high northern latitudes at an altitude of 25–35 km (where streamers are to be expected) in 2017. That was not clear enough, and the term 'mid-stratosphere' is open to interpretation. The paragraph has been slightly modified in several places, with the most important changes below:

"During the OMPS period from 2012 to the present, there were several volcanic eruptions that had an impact on the stratosphere (e.g. Carn, 2025; Kovilakam et al., 2025). (...) For the altitudes and latitude regions shown in Fig. 1, only a direct impact from the Hunga eruption is to be expected, whose plume was observed at an altitude of approximately 26 km (Duchamp et al., 2023). (...) The other eruptions referenced are not expected to exert a direct significant influence in the Northern Hemisphere on the altitudes shown."

**References:**

Carn, S.: Multi-Satellite Volcanic Sulfur Dioxide L4 Long-Term Global Database V4, https://doi.org/10.5067/MEASURES/SO2/DATA405, 2025.

Kovilakam, M., Thomason, L. W., Verkerk, M., Aubry, T., and Knepp, T. N.: OMPS-LP aerosol extinction coefficients and their applicability in GloSSAC, Atmospheric Chemistry and Physics, 25, 535–553, <a href="https://doi.org/10.5194/acp-25-535-2025">https://doi.org/10.5194/acp-25-535-2025</a>, 2025.

Lastly on this point, it would be useful to acknowledge that Hunga was a peculiar eruption with a large stratospheric moistening, and discuss what this means for the air parcel and for the assumptions beyond the PSD.

The interaction of water vapour and aerosols and their possible influence on streamers and FrIACs in 2022 and 2023 are certainly interesting. However, apart from this mention, the Hunga eruption plays no role in the further analysis of the streamers and anticyclones described in this paper, and the question lies outside the scope of the study.

Fig. 3 and Fig. 4 and Fig. 7, 8: The colors in these two figures are essentially impossible to understand when overlaid with the gray shading of the continents. Please only leave the countours of the continents and make the color constrat a bit sharper, or use black contour lines like the third row of Fig. 4.

We increased the figure size of Figs. 4 and 8. The color scale in Figs. 3 and 7 has now been adjusted and the color scheme of O3 has been changed in Figs. 4 and 8. The continents are no longer filled.

**L. 310: not sure "bins" is the right word here. Gridboxes?**

Grid boxes sounds more appropriate. Changed.

L. 321: with all due respect to the Grainger PDF, which I've used often as well, as a reference with no permanent identifier is not really suitable for a journal. The internet often forgets...

The reference to the Grainger PDF has been replaced by a reference to Seinfeld and Pandis 2016. A mention of the Grainger PDF has been moved to the Acknowledgements.

**References:**

Seinfeld, J. H. and Pandis, S. N.: Atmospheric Chemistry and Physics: From Air Pollution to Climate Change, New York Academy of Sciences Series, John Wiley & Sons, Incorporated, Newark, 1st ed edn., ISBN 978-1-118-94740-1 978-1-119-22116-6, 2016.

Finally, I found the Conclusions particularly brief and dry. It seems to include some points that are more suitable for the Discussion section, and is lacking a final section that discusses future directions and an explanation of what this research contributes.

You're right, the conclusions section wasn't well formulated yet. The conclusions have been completely revised and the last paragraph now reads:

"The transport of tropical aerosol-rich air to mid-latitudes is not an uncommon phenomenon and occurs to a small extent multiple times every year. However, the observation of a tropical-subtropical streamer reaching high latitudes is a rare occurrence. In 2017, this is even evident in the zonal mean aerosol extinction at high latitudes. The correlation between streamer events and aerosol extinction coefficients was demonstrated for the first time in this study. This correlation as well as the connection between streamers and anticyclones in 2017 provide new insights into the poleward transport and persistence of tropical air in the middle stratosphere at the end of the polar vortex season."

**References**

Li, Y., Pedersen, C., Dykema, J., Vernier, J.-P., Vattioni, S., Pandit, A. K., Stenke, A., Asher, E., Thornberry, T., Todt, M. A., Bui, T. P., Dean-Day, J., and Keutsch, F. N.: In situ measurements of perturbations to stratospheric aerosol and modeled ozone and radiative impacts following the 2021 La Soufrière eruption, Atmos. Chem. Phys., 23, 15351–15364, https://doi.org/10.5194/acp-23-15351-2023, 2023.

---

## Author Comment (AC3)

**Replies to comments by Farahnaz Khosrawi**

Comments, replies, "changes in the manuscript"

References to added publications not already cited in the first version of the paper are listed in the replies.

I have read this manuscript with great interest. The authors present important new results that deserve to be published. However, I was quite disappointed about the presentation of the results (in terms of quality of the figures) and the text itself (too confusing, in some parts too lengthy and many statements without providing references). I will provide detailed comments below.

Thank you very much for your very detailed and helpful comments. We think that the suggestions for improvements to the text and figures have enhanced the presentation of the results and the clarity of the argumentation.

**Major comments:**

Introduction: 3.5 pages for an article introduction is much too long. I would suggest to shorten the introduction to 1.5-2 pages.

You are right, the introduction was too long and unfocused. We have shortened all the paragraphs a little, as well as removing some and merging others. At 2.5 pages, the introduction is still relatively long, but we think it is within reasonable limits.

Figures: It would really be worth to increase all figure sizes, especially for the figures which consist of several panels and stereographic projections (Fig. 3, 4, 7 and 8). I also had trouble with the color scale. Therefore, I would suggest also to improve/change the color scales for some of them.

We increased the figure size of Figs. 3, 4, 7 and 8. The color scale in Figs. 3 and 7 has now been adjusted and the color scheme of O3 has been changed in Figs. 4 and 8. The continents are no longer filled.

Discussion: In my opinion it would also be worth to shorten the discussion. This section was also too lengthy and in some parts confusing since it was not clear why specific things are discussed. See also my specific comments below.

As explained in more detail later in the specific comments, we have omitted or shortened some paragraphs in the discussion.

**Section 5: This section rather belongs to the method section or to an appendix.**

We have discussed this and, in our opinion, this section does not belong in the methodology section, as it is more of an extension of the discussion regarding errors. Furthermore, we would like to keep this section in the main body of the paper. As suggested by Anonymous Referee #2, we have now moved section 5 before the discussion.

**Specific comments:**

P2, L41: Dividing the atmosphere into three regions is useful only not only for understanding transport processes in the polar stratosphere, but also for understanding transport processes in the entire stratosphere.

P2, L41ff: Mention also how Plumb (2002) came up with the idea of dividing the stratosphere into three regions. Which method did he use? PDFs or tracer-tracer correlations?

Lines 41ff have been changed to:

"In order to understand the transport processes in the stratosphere, Plumb (2002) suggests dividing it into three meridional regions: the tropics, the surf zone and the polar vortex. To this end, Plumb examined the relationships between long-lived tracers and reference tracers in these regions."

P2, L54 or L70: Here, you could also add the reference of Khosrawi et al. (2005) who investigated the development of a streamer as measured by CRISTA and modeled with CLaMS and KASIMA. Also the study of Eyring et al. (2003) could be worth to be cited.

We had overlooked both publications before, thank you very much for pointing them out. References to Eyring et al. 2003 and Khosrawi et al. 2005 have been added in L70.

P2, L55ff: Add "e.g." before the given reference since there are much more and you are providing here only some examples.

Done.

P3, L67-72: Add here some more newer references, too.

References to Kouker et al. 1999, Eyring et al. 2003, Khosrawi et al. 2005 and Wüst et al. 2025 have been added.

**References:**

Kouker, W., Offermann, D., Küll, V., Reddmann, T., Ruhnke, R., and Franzen, A.: Streamers observed by the CRISTA experiment and simulated in the KASIMA model, Journal of Geophysical Research: Atmospheres, 104, 16 405–16 418, <a href="https://doi.org/10.1029/1999JD900177,1999">https://doi.org/10.1029/1999JD900177,1999</a>.

Wüst, S., Küchelbacher, L., Trinkl, F., and Bittner, M.: Gravity waves above the northern Atlantic and Europe during streamer events using Aeolus, Atmospheric Measurement Techniques, 18, 1591–1607, <a href="https://doi.org/10.5194/amt-18-1591-2025">https://doi.org/10.5194/amt-18-1591-2025</a>, 2025.

P3, L80f: References? I guess the stratospheric aerosol distribution in the vortex has previously been documented in the literature.

As a reference for the aerosol distribution in the vortex, Harvey et al. 1999 is now cited in the penultimate paragraph of the introduction alongside Thomason et al. 2007.

**References:**

Harvey, V. L., Hitchman, M. H., Pierce, R. B., and Fairlie, T. D.: Tropical aerosol in the Aleutian High, Journal of Geophysical Research: Atmospheres, 104, 6281–6290, <a href="https://doi.org/10.1029/1998JD200094">https://doi.org/10.1029/1998JD200094</a>, 1999.

P3, L80: The connection between filaments in trace gas distribution and in aerosol distribution is not clear. This paragraph needs a better transition.

The transition to satellite measurements and aerosols now reads:

"The observability of streamers and FrIACs has not yet been demonstrated for aerosols."

**P4, L120-122: Don't mention here each subsection, just summarize what is presented in Sect. 3.**

The paragraph now reads:

"Section 3 describes the evolution of the aerosol extinction coefficient in the northern hemisphere in spring 2017 and 2005 and compares it with meteorological reanalysis data."

**P5, L127ff: This paragraph rather belongs to the introduction.**

As the introduction is already quite lengthy, we have decided to leave the choice of instruments and the explanation of the methodology of limb scattering instruments in the Methodology section.

**P5, L154: The first sentence of this paragraph is rather obsolete.**

Changed. The first sentence has now been omitted and the second sentence changed to:

"One advantage of the OMPS-LP data product from the University of Bremen is that the aerosol extinction coefficient is usually derived to a greater height than the derived aerosol product of NASA."

**P6, L184: What do you do to overcome this? Is the affected SAGE III data still usable?**

The following sentence was added at the end of the paragraph:

"Therefore, the robustness of the derived PSD also decreases with altitude."

In section 5, the effects were tested when the PSD is not used across the entire altitude range, but rather when averaged at an altitude of 25–30 km. It results only in a deviation of about 5 % in aerosol mass.

**P6, L186: I would suggest to rephrase this sentence to be more clear. The ERA5 data itself is also data of a model simulation where observations have been assimilated into.**

The entire chapter has been revised. The first sentence has been slightly changed:

"The ECMWF ERA5 meteorological reanalysis data is used for this study to investigate the atmospheric dynamics in specific periods (Hersbach et al. 2020)."

In addition, a paragraph on the validation of stratospheric transport in ERA5 has been added.

**P7, L195: Add the time period.**

**P7, L198: Add references that document these volcanic eruptions.**

The entire paragraph has been slightly modified and the first line now reads:

"During the OMPS period from 2012 to the present, there were several volcanic eruptions that had an impact on the stratosphere (e.g. Carn, 2025; Kovilakam et al., 2025)

**References:**

Carn, S.: Multi-Satellite Volcanic Sulfur Dioxide L4 Long-Term Global Database V4, <a href="https://doi.org/10.5067/MEASURES/SO2/DATA405">https://doi.org/10.5067/MEASURES/SO2/DATA405</a>, 2025.

Kovilakam, M., Thomason, L. W., Verkerk, M., Aubry, T., and Knepp, T. N.: OMPS-LP aerosol extinction coefficients and their applicability *in GloSSAC*, *Atmospheric Chemistry and Physics*, *25*, 535–553, <a href="https://doi.org/10.5194/acp-25-535-2025">https://doi.org/10.5194/acp-25-535-2025</a>, *2025*.

**P8, L204: When exactly was the maximum reached? Add the year or exact time period.**

**Changed:**

"After the eruption, the aerosol extinction coefficient reaches its maximum at an altitude of 25.5 km in the northern subtropics during the second half of 2022."

**P8, L216: Why was there a higher background aerosol level? Has this been documented in the literature? References?**

This is only a description of what we see at this altitude and latitude range in Fig. 1. The extinction coefficient curves in Fig. 1 show a great deal of natural variation. Interpreting all these fluctuations is beyond the scope of this paper. The sentence has been slightly modified:

"At an altitude of 28.5 km, the background level of the aerosol extinction coefficient is higher in other years than during early 2017 due to natural variations."

**P8, L221-224: First, you mention less pronounced peaks and then a twofold increase. How does this fit together?**

**Changed to:**

"In the northern mid-latitudes, indicated by the red line in Fig. 1, two distinct peaks can be identified at altitudes of 28.5 km, 31.5 km and 34.5 km in March and April 2017, which are slightly less pronounced than in the high latitudes."

P9, Figure 2: I was wondering about your marking of the SSWs. I would have them expected during other times looking at this graph. While doing a quick search in the internet I found the paper by Roy and Kuttipurath (2022) where it is stated that the warmings in 2016/217 were in early January and early February.

We cannot find this statement in Roy and Kuttipurath (2022). From their Fig. 1, two minor SSWs can be deduced in February 2017. Their sentence on page 4:

"The temperature increased from 206 to 230 K (…) at  $60^{\circ}$  N during early January to early February in 2017 (…)."

refers to the development of the first minor warming in February. E.g. Eswaraiah et al. 2019, Gogoi et al. 2023 and Li et al. 2023 all identify two minor warmings in winter 2016/17, one at the end of January/beginning of February and one at the end of February. The references to the publications have now been added to the text and L227f has been slightly modified:

"Two events, which are classified as minor warmings, are recorded at the beginning and end of February, which is consistent with the observations in Eswaraiah et al. 2019, Gogoi et al. 2023 and Li et al. 2023."

**References:**

Eswaraiah, S., Venkat Ratnam, M., Kim, Y. H., Kumar, K. N., Venkata Chalapathi, G., Ramanajaneyulu, L., Rao, S. V. B.: Advanced meteor radar observations of mesospheric dynamics

during 2017 minor SSW over the tropical region. Adv. Space Res., 64(10), 1940–1947. https://doi.org/10.1016/j.asr.2019.05.039, 2019.

Gogoi, J., Bhuyan, K., Sharma, S. K., Kalita, B. R., & Vaishnav, R.: A comprehensive investigation of Sudden Stratospheric Warming (SSW) events and upper atmospheric signatures associated with them. Adv. Space Res., 71(8), 3357–3372. <a href="https://doi.org/10.1016/j.asr.2022.12.003">https://doi.org/10.1016/j.asr.2022.12.003</a>, 2023.

Li, Y., Kirchengast, G., Schwaerz, M., & Yuan, Y.: Monitoring sudden stratospheric warmings under climate change since 1980 based on reanalysis data verified by radio occultation. Atmos. Chem. Phys., 23(2), 1259–1284. <a href="https://doi.org/10.5194/acp-23-1259-2023">https://doi.org/10.5194/acp-23-1259-2023</a>, 2023.

P10, Figure 3: Increase the figure size and adjust color scale so that the differences in the aerosol distribution between the considered time periods become more obvious.

The figure size has been increased, the colour scale has been adjusted and the continents are no longer filled.

P11, Figure 4: Please increase the figure size and use a different color schemes than the current ones for PV and O3. Add also the time periods considered in the figure caption (or write a note that this are given in the plot title).

We have increased the figure size, changed the color scheme of O3, and the continents are no longer filled. In our opinion, the signatures in PV can be recognised using the selected colour scheme, which is also CVD-friendly. We have therefore decided to retain the colour scheme for PV. The following sentence has been added to the captions of Figures 4 and 8:

"(…), indicated above the first row."

P12, 272: The transport barrier is between "air masses" not "air parcels".

P12, in general: I had trouble to understand how you can investigate mixing processes while considering profiles. This needs more explanations and guidance for the reader.

The term 'transport barrier' was used incorrectly in this context. The sentence has been adjusted and now appears later in the same paragraph.

We think that the comprehension issues arose from the ambiguous use of 'mixing' and 'transport barrier'. The idea behind this approach is that two profiles of the aerosol extinction coefficient at the start of the streamer event and three weeks later, which are very similar, are shown. From this we conclude that there has been little mixing with the ambient air along the path. The paragraph now reads:

"At an altitude of 26–38 km the vertical profiles in the tropics and three weeks later in the anticyclone differ only slightly. This suggests that there was little mixing of aerosols with the ambient air along the transport path. The high-pressure area, which transports tropical air towards higher latitudes, creates a barrier for mixing between the anticyclone and the surrounding air."

P12, 275-276: "The agreement between anticyclone and subtropics....." This does not make any sense for me.

Changed to:

"The agreement between the profiles inside the anticyclone and from the subtropics(…)"

P12, L282: Refer to the respective figure or add a reference.

We considered adding further years with signatures from streamers or FrIACs to the appendix. However, these are not self-explanatory and require an explanation and discussion of the atmospheric conditions. We therefore decided to leave it at the description and presentation of the cases in 2017 and 2005. The additional examples mentioned now have '(not shown)' added after them.

**P12, L285: This is not visible for me, even when I zoom into the figure.**

The image has now been enlarged. We decided against adding additional markings to the plots in order to avoid further confusion caused by additional symbols and colours.

The paragraph has been adjusted to provide a clearer indication of where to look and to correct the statement:

"As of 13 March, a weak polar vortex streamer becomes discernible in the reanalysis data, see the second column of Fig. 4. This is evident below 45° N over southern Europe in an increased potential vorticity, see Fig. 4j, an increased specific humidity, see Fig. 4N, and also a reduced aerosol extinction coefficient, see Fig. 4b."

**P14, L307ff: This paragraph rather belongs to the method section.**

You are of course right that this section mainly contains methods. Nevertheless, we have decided not to split this section and to leave it in the results so as not to anticipate and to maintain the line of argumentation.

**P14, L327: How do you know that the mixing barrier was strong?**

The term 'mixing barrier' was used incorrectly here. The sentence has been changed to:

"Since the aerosol extinction coefficient changes only slightly along the transport path of the streamer, see Fig. 5, it is assumed that changes in mass along the path are also negligible."

**P14, L333: Why are you suddenly in this section focus on 2005? A motivation or transition between the sections is missing.**

Added the following sentences:

"Manney et al. 2006 observed a strong and long-lasting FrIAC in 2005. This subsection will examine whether this can also be observed in the aerosol extinction coefficient."

**P15, Figure 7: Increase figure size and improve color scale.**

The figure has been increased, the color scale improved and the continents are no longer filled.

**P15, L345: Add the date and a reference.**

A reference to Butler & Domeisen 2021 for the date and historical median of the final warming has now been added in line 338f.

**References:**

Butler, A. H. and Domeisen, D. I. V.: The wave geometry of final stratospheric warming events, Weather Clim. Dyn., 2, 453–474, https://doi.org/10.5194/wcd-2-453-2021, 2021.

**P16, Figure 8: Increase figure size and use another color scheme for PV and O3.**

We have increased the figure size, changed the color scheme of O3, and the continents are no longer filled. In our opinion, the signatures in PV can be recognised using the selected colour scheme, which is also CVD-friendly. We have therefore decided to retain the colour scheme for PV.

P17, L360: To my knowledge, filaments do not necessarily occur in connection with anticyclones. I may be wrong, but this statement should be better elaborated.

That's right, streamers and anticyclones are not inherently related to each other. The sentence was changed as follows:

"The duration and stability of a streamer depend on the dynamics in mid-latitudes in connection with the polar vortex and whether the streamer is associated with an anticyclone, which, as in 2017, prevents mixing with the surrounding air."

P17, L361: Please check the dates for warmings again. See Roy and Kuttipurath (2022) who write that these were in early January and early February.

As before, we cannot find this statement in Roy and Kuttipurath 2022.

P17, L380: What is the connection between tropical-subtropical streamers and the polar vortex? To my knowledge streamers can also occur when there is no polar vortex (thus in other seasons than solely in winter time).

According to Offermann et al. 1999 and Eyring et al. 2003, the streamer frequency is significantly higher in winter than in summer. This line and similar statements have therefore been corrected:

"Tropical-subtropical streamers can occur at any time of the year, although streamer frequency is significantly higher during the polar vortex season (Offermann et al., 1999; Eyring et al., 2003)."

**P17, L384: Add a reference.**

A reference to van Loon et al. 1973 has been added.

P18, "observed signal"? Which exactly? Which parameter? Which figure are you referring to? Your study is using a completely different approach than previous studies, how can these then be compared?

Two mentions of the word 'signal' on this page have now been specified:

L. 391: "The observed signal of the instrument (...)"

L.394: "The streamer signatures observed in the aerosol extinction coefficients in 2017 show good agreement (...)"

**P18, L401: Which model?**

The FUB-CMAM model used in the referenced study by Krüger et al. 2005. This is now emphasised more strongly:

L. 394: "(...) expected from runs for stratospheric streamers with the Freie Universität Berlin Climate Middle Atmosphere Model (FUB-CMAM) in Krüger et al. 2005."

L. 400: "(...) also occurs in the model used in Krüger et al. 2005 in some years."

P18, L411: Add a reference. For me it is not clear how QBO phase is related to the results presented in your study. Why are you discussing this?

This was discussed because the QBO phase has an influence on the stability of the polar vortex and on the probability of SSWs. The paragraph has now been omitted.

P17-19: The discussion is too lengthy and to confusing and should be shortened. Focus only on the important points and discuss these.

The paragraph on QBO has now been omitted and a few paragraphs have been merged. In our opinion, the length of the discussion is now reasonable.

P19, Section 5: This section rather belongs to the method section or should be in an appendix.

In our opinion, this section does not fit into the methods section, and we would like to keep it in the main body of the paper. As suggested by Anonymous Referee #2, we have now moved section 5 before the discussion.

P20, L485: Isn't this a contradiction? If you have a streamer from the tropics to the subtropics, how is this then related to the polar region and the polar vortex?

The term tropical-subtropical streamers (used as in Offermann et al. 1999 and Krüger et al. 2005) refers to the origin of the streamers.

P20, L465: In which direction is the flow of the streamer? From high to low latitudes or from low to high latitudes?

As emphasised in line 461, the tropical-subtropical streamers mentioned in this study cause transport from low to high latitudes. This study contains only two references to polar vortex streamers, which transport air masses from high to lower latitudes, in lines 57 and 283.

**Technical corrections:**

P2, L46 and 47: For better readability, instead of "it" it should be clearly mentioned what is meant.

Changed to:

"The edge of the polar vortex can be characterized by (...)".

P3, L80: Add this sentence to the previous paragraph.

The first sentence of this paragraph has now been removed and the second sentence added to the following paragraph.

P5, L144: remove comma and add "as".

Done.

P6, L172: Instead of "several times" I would suggest to write "previously" or "in other studies".

Changed to "previously".

P6, L186: Add "meteorological" before "reanalyses".

Done.

P8, L208: Add this sentence to the previous paragraph.

Done.

P10, L256: Chose a different term than "material". Just name what exactly it is, either the respective trace gas or the aerosols.

P10, L258 and 259: Don't write "it", clearly state what exactly.

Two sentences in this paragraph have been changed:

"As of 16 March, the aerosols are observed to depart from the edge of the polar vortex. (...) The aerosols are kept at the same longitude in the anticyclone for about three weeks, while slowly moving towards lower latitudes."

P12, L282: Add this sentence to the previous paragraph.

Done.

P14, L337: Add "e.g." before the reference of Sonkaew et al.

Done.

P17, L372: Chose another term than "material".

Changed to "aerosols".

P17, L382: "structure's trajectory" I would suggest to rephrase this.

The sentence has been changed:

"However, the coverage of the high latitudes by the limb instruments used decreases towards the end of the year, making it more difficult to track the course of a streamer."

P19, L426-428: Add these sentences to the previous paragraph.

Done.

P20, L470ff: Combine these sentences to one paragraph instead of several small paragraphs.

The conclusions were revised and the paragraphs rearranged.

**References:**

Eyring, V., Dameris, M., Grewe, V., Langbein, I., and Kouker, W.: Climatologies of subtropical mixing derived from 3D models, Atmos. Chem. Phys, 3, 1007–1021, https://doi.org/10.5194/acp-3-1007-2003, 2003.

Khosrawi, F., Grooß, J.-U., Müller, R., Konopka, P., Kouker, W., Ruhnke, R., Reddmann, T., and Riese, M.: Intercomparison between Lagrangian and Eulerian simulations of the development of mid-latitude streamers as observed by CRISTA, Atmos. Chem. Phys., 5, 85–95, https://doi.org/10.5194/acp-5-85-2005, 2005.

Roy, R. and Kuttipurath, J., The dynamical evolution of Sudden Stratospheric Warmings of the Arctic winters in the past decade 2011–2021, SN Applied Sciences, https://doi.org/10.1007/s42452-022-04983-4, 2022.

---

## Author Comment (AC4)

**Replies to comments by Referee 2**

Comments, replies, "changes in the manuscript"

References to added publications not already cited in the first version of the paper are listed in the replies.

This article presents satellite observations of stratospheric steamers and frozen-in anticyclones in the enhancement of aerosol extinction at mid and high latitudes. This is the first study to document these phenomena using observations of stratospheric aerosols. The article demonstrates that these events can contribute significantly to the transport of sulphuric acid aerosols to the stratosphere at mid- and high-latitudes. The article is well-written and well documented. It provides new information on stratospheric air transport between tropical and higher latitudes following major SSWs. It is worthy of publication in Egusphere after a few minor revisions, as detailed below.

Thank you very much for the helpful comments and suggestions.

Line 32: The definition of major SSWs by the reversal of zonal wind at 60°N-10 hPa was first proposed by Labitzke (1981), well before Charlton and Polvani (2007).

There were also other SSW definitions prior to Labitzke 1981, and they all differ slightly from Charlton and Polvani 2007. Labitzke 1981 refers to a WMO CAS definition, which was slightly modified in her study.

For our study, Labitzke 1981 is referenced in relation to the definition of minor SSWs. For major SSWs, we opted for the simplified definition according to Charlton and Polvani 2007, which only includes the criterion of the reversal of the zonal-mean zonal winds at 10 hPa and 60° latitude. It is frequently applied, and no wind reversal due to major SSWs can be observed in the two years under consideration.

Line 165: The phrase "vertical resolution greater than 7 km" may be confusing. I suggest "vertical resolution better than 7 km".

You are right this phrase was misleading. The sentence now reads:

"For both data sets, only retrieved profiles with a vertical resolution better than 7 km were used to ensure high data quality at all altitudes."

Lines 180-184: It is true that the possibility of negative values for extinction can lead to a negative mean value, but conversely, the exclusion of negative values can lead to a positive bias in the mean value.

That is correct. This paragraph should serve as a general reference to the problems of SAGE III retrieval above  $\sim$ 32 km, especially at small wavelengths. At the 449 nm channel, approximately two-thirds of the measurements at an altitude of 35 km are negative and one-third are positive, whereas one would expect a maximum of 50% negative values as the altitude increases and the signal strength decreases.

However, the solution algorithm for the PSD only works if the extinction coefficient is positive in all channels used in the algorithm (Knepp et al. 2024).

In the subsection, a few sentences were rearranged and the following information was added:

"Their solution algorithm for the PSD only works if the extinction coefficient is positive in all channels used in the algorithm.

"Therefore, the robustness of the derived PSD also decreases with altitude."

**Figure 2: The zonal wind reaches a value close to 0 for the SSW event occurring at the end of February. I wonder if it should be classified as a major SSW.**

It is close, but since there is no reversal of the zonal wind, we have classified this event as a minor SSW, as have Eswaraiah et al. 2019, Gogoi et al. 2023 and Li et al. 2023. The following sentence has been added to the text:

"Two events, which are classified as minor warmings, are recorded at the beginning and end of February, which is consistent with the observations in Eswaraiah et al. 2019, Gogoi et al. 2023 and Li et al. 2023."

Section 5: Sources of error: it would be preferable to place this section before the Discussion section.

Thank you for the suggestion, we have now moved section 5 before the discussion.

Line 468-469: the sentence on the non-significant contribution of increased aerosol on the radiative transfer in the conclusions is not supported by any results in the text of the paper. It should be removed from the conclusions or supported by some argumentation in the core of the paper.

This sentence has now been removed and the conclusions section revised.

---

## Author Response (AR2)

**Replies to Report #1 by Referee 3 Farahnaz Khosrawi**

**Comments,** replies**,** *„changes in the manuscript"*

**appreciate that the authors have taken my comments into account and revised their manuscript accordingly. Before publication I would like to ask the authors to consider the following technical corrections:**

Thank you very much for your detailed review of both versions. We believe it has improved the paper.

**- P1, L3: ratio -> ratios**

Changed.

**- P2, L23: What is meant with "frequently occuring zonal indicator"? Please revise.**

This sentence has been revised:

*"So-called sudden stratospheric warmings (SSWs) are frequently occurring disturbances."*

**- P2, L34: instead of "it" write clearly "the stratosphere" or write " suggest a devision into three...." to avoid repetition of "stratosphere" in this sentence.**

Changed to:

*"suggests a division into three meridional regions: (…)."*

**- P2, L26: Here it needs to be shortly explained what the surf zone is.**

**- P2, L39: Latest here it should be explained what the surf zone is (see my comment on L26).**

Added *"so-called"* before surf zone in L26. L37-L39 has now been expanded:

*"As described in McIntyre and Palmer (1984), the surf zone is the region in the mid-latitudes into which air masses can be transported vertically by breaking planetary waves, resulting in extensive mixing of trace gases."*

**- P3, L79: Mention/explain what the Junge layer is. At least the altitude where the Junge layer is found should be added.**

The term "Junge layer" has now been replaced by *"stratospheric aerosol layer"* and the altitude region has been added.

*"The extinction coefficient above the stratospheric aerosol layer (roughly at 15–25 km altitude) (…)."*

**- P3; L83: Here you should clearly state that the dynamic effects and meteoric or space debris contribute to particle formation. Otherwise the sentence is not clear and rather misleading. You could simply write "particle formation due to dynamic effects, meteoric or space, debris".**

Stratified dynamic effects do not cause particle formation, but rather the possible observability of a transported air mass with increased aerosol content compared to the air masses below. The statement was therefore expanded as follows:

*"An increasing extinction coefficient with altitude can therefore be explained either by stratified dynamic effects, in which a transported air mass is observed that is richer in aerosols than the air below, or by material coming from above, such as meteoric dust or space debris."*

**- P3, L85: Also here for clarity a short explanation what the subduction zone is would be useful.**

The mention of the subduction zone has now been omitted.

**- P3, L103: add "instrument" so that it reads "remote sensing instruments".**

Done.

**- P5, L130: height -> altitude**

Done.

**- P6, L181: Add this sentence to the next paragraph to avoid having a paragraph consisting of one sentence.**

Done.

**- P6, L188 and throughout the manuscript: Omit space between figure number and panel label, e.g. Fig. 1 e should read Fig. 1e.**

We had added spacing between the figure number and panel label, as it looked confusing with some figure labels (e.g. 4l). We will leave the spacing as it is for now and leave this decision to the typesetting.

**- P7, Fig 1 caption and throughout the manuscript: Check the Copernicus guidelines, if I remember it correctly northern and southern hemisphere is written beginning with small letters.**

According to Collins and Merriam-Webster, both spellings are acceptable. When looking through published papers, the spelling with capital letters seems to be more common. We have therefore decided to retain this spelling.

**- P10, L245: it -> be more clear. What exactly?**

Replaced with *"the anticyclone"*.

**- P10, L252: "Fig." before figure number missing. It should read "Fig. 4l".**

Done.

**- P12, Fgure 5 caption: omit "0" and just write "4" and "5" March.**

**- P13, Figure 6 caption: same here as fo P12 Figure 5, omit the "0".**

"0" has now been omitted from the captions of Figs. 5 and 6, as well as from Fig. 6.

**- P13, L292: Has the abbreviation "PSD" been introduced? If the term is only used for a few occasion I would suggest to rather skip using this abbreviation for better readability.**

The abbreviation PSD for particle size distribution was introduced on page 3, line 77 and is used nineteen times throughout the text. We have decided to retain this abbreviation.

**- P18, L373: depend -> depends**

"The duration and stability of a streamer depend on the dynamics in mid-latitudes...". Since the verb refers to two things, we think that "depend" is correct.

**- P19, L425: Add also respective altitude in km.**

The respective altitude has now been added in three places in this paragraph in lines 420 and 427.

L420:*"(10 hPa, ∼ 31 km)...(20–30 hPa, ∼ 23–26 km)"*

L427:*"(5–20 hPa, ∼ 26–35 km)"*

**- P19, L436: Omit "of the atmsopheric state".**

Done.

**- P20, L444: Here zyou write "sulphuric", before you wrote "sulfuric". Use a consistent writing.**

Changed to *"sulfuric"*.

**Reference list: The reference of Khosrawi is imcomplete. The co-authors as well as the journal number and doi are missing.**

We apologise for this oversight. The complete references for Khosrawi et al. 2005, as well as for Thomason et al. 2007 and Thomason et al. 2010, have now been added.

**Replies to Report #2 by Referee 1**

**I think the authors have done a very good job responding to the reviews. I especially appreciate the new streamlined Introduction, the higher quality figures and the clearer, crisper conclusions. I recommend publication and thank the authors for their work!**

Thank you very much for the review, the feedback and the kind words.